# High-Flow Nasal Cannula Oxygen Therapy versus Non-Invasive Ventilation in patients at very high risk for extubating failure: A systematic review of randomized controlled trials

**Ziyad F. Al Nufaiei** [1,2,3]*, **Raid M. Al Zhranei**[1,2,3]

**1** College of Applied Medical Sciences, King Saud Bin Abdulaziz University for Health Sciences, Jeddah, Saudi Arabia, **2** King Abdullah International Medical Research Center, Jeddah, Saudi Arabia, **3** Ministry of the National Guard—Health Affairs, Jeddah, Saudi Arabia

* zalnufaie@gmail.com

## Abstract

### Background

Mechanical ventilation is commonly used for managing respiratory failure in chronic obstructive pulmonary disease (COPD) patients, but weaning patients off ventilator support can be challenging and associated with complications. While many patients respond well to Non-invasive ventilation (NIV), a significant proportion may not respond as favourably. We aimed to assess whether high-flow nasal cannula (HFNC) is equally effective as NIV in reducing extubation failure among previously intubated COPD patients.

### Methods

This systematic review was carried out in line with PRISMA guidelines We searched PubMed, Scopus, Web of Science, and Cochrane library from inception until February 15, 2023. Randomized Clinical Trials (RCTs) of adults at high risk for extubating failure were included. We examined the use of HFNC as the intervention and NIV as the comparator. Our outcome of interest included, reintubation rate, length of hospital or intensive care unit (ICU) stay, adverse events, and time to reintubation. The Cochrane risk-of-bias tool was used for randomized trials to assess risk of bias.

### Results

We identified 348 citations, 11 of which were included, representing 2,666 patients. The trials indicate that HFNC is comparable to NIV in preventing reintubation after extubating in COPD patients. In comparison to NIV, HFNC also produced improved tolerance, comfort, and less complications such as airway care interventions. NIV with active humification may be more effective that HFNC in avoiding reintubation in patients who are at extremely high risk for extubating failure.

**Data Availability Statement:** All relevant data are within the manuscript and its Supporting Information files.

**Funding:** The authors received no specific funding for this work.

**Competing interests:** The authors have declared that no competing interests exist.

## Conclusion

The inconclusive nature of emerging evidence highlights the need for additional studies to establish the efficacy and suitability of HFNC as an alternative to NIV for previously intubated COPD patients. Clinicians should consider the available options and individualize their approach based on patient characteristics. Future research should focus on addressing these gaps in knowledge to guide clinical decision-making and optimize outcomes for this patient population.

## Introduction

Mechanical ventilation is a widely used therapeutic approach for managing respiratory failure, particularly in chronic obstructive pulmonary disease (COPD) patients experiencing acute exacerbation [1–3]. However, the process of weaning patients from ventilator support poses a significant challenge, often leading to ventilator fatigue and deteriorating hemodynamics [1]. Prolonged invasive mechanical ventilation is associated with increased risks of complications, such as ventilator-associated pneumonia, and difficulties in successfully removing patients from ventilation [4]. Non-invasive ventilation (NIV) has shown promise in facilitating the transition from invasive ventilation to independent breathing for individuals with COPD, resulting in improved outcomes and prognosis [5, 6]. However, a subset of patients, approximately 15% to 25%, does not respond favorably to NIV, primarily due to low tolerance, leading to the need for endotracheal intubation [7, 8]. Consequently, there is a pressing need for alternative respiratory support methods for COPD patients who cannot tolerate or have contraindications to NIV after extubation. Additionally, NIV has been proven effective in providing stable high-concentration oxygen, improving lung function, enhancing gas exchange, and reducing the need for intubation and mortality in patients with respiratory failure, particularly in those with exacerbated COPD [9]. However, the use of NIV is associated with various side effects, including skin injury, eye irritation, claustrophobia, dryness of the mouth and throat, flatulence, and an increased risk of aspiration, which limit its clinical utility [10].

High-flow nasal cannula (HFNC) oxygen therapy has emerged as a promising alternative respiratory support system. HFNC delivers a high flow of heated and humidified gas through specialized nasal prongs, promoting positive airway pressure, enhancing functional residual capacity, optimizing oxygenation, and providing continuous oxygen concentration. The humidification of the gas facilitates airway fluid secretion and clearance, protects airway epithelial cells, and improves patient comfort during treatment [11–13]. Previous studies have reported that HFNC is associated with greater patient comfort and tolerability compared to NIV [14, 15]. However, limited research has investigated the effectiveness of HFNC in reducing extubation failure among high risk COPD patients. Therefore, this systematic review aimed to evaluate whether HFNC is as effective as NIV in reducing extubation failure in previously intubated COPD patients.

## Methods

This systematic review followed the Preferred Reporting Items for Systematic Reviews and Meta-Analysis (PRISMA) statement [16]. The methods were guided by the Cochrane Handbook of Systematic Reviews and Meta-analysis of Interventions (version 5.1.0).

### Inclusion criteria

We utilized the PICO framework (Population, Intervention, Comparator, and Outcome), to guide our criteria selection. We included randomized controlled trials (RCTs) that focused on COPD patients at very high risk for extubating failure and examined the use of HFNC therapy as the intervention. These studies were required to compare HFNC Therapy with NIV as the comparator. Our outcome of interest encompassed variables such as reintubation rate, length of stay, adverse events, and time to reintubation.

We excluded RCTs that were not in English, and any other study designs (e.g., observational studies).

### Search strategy

We performed a comprehensive literature search on four electronic databases (PubMed, Scopus, Web of science, and Cochrane library) from inception until February 15, 2023. We also searched reference lists of all included studies. **S1 Appendix** details the search terms used and PICO framework.

### Screening, data extraction and quality assessment

Two authors (ZN, RZ) independently screened titles, abstracts and full texts of all identified records for eligibility using Endnote. Data were extracted independently by two authors (ZN, RZ), including citation details, study characteristics, participant characteristics relevant to the selection criteria, risk of bias domains, and outcome measures, such as, reintubation rate, length of stay, mortality, adverse events, and time to reintubation. We independently assessed the quality of included RCTs using The Cochrane Risk of Bias 2 (RoB 2) tool [17]. It involves the following five domains: randomization process (selection bias), deviation from intended interventions (performance bias), outcome measurement (detection bias), missing outcome data (attrition bias), selection of reported results (reporting bias) and other potential sources of bias. We classified the studies as 'Low risk of bias', 'High risk of bias" or 'Some concerns". Discrepancies were resolved by discussion between the authors. To explore the publication bias across studies, we constructed funnel plots to present the relationship between effect size and standard error.

## Results

A total of 348 citations were identified from the database searched. Of which, 107 were duplicates and were removed. A further 151 articles were excluded after screening titles and abstracts. Another 30 articles were removed as they were not matching our inclusion criteria. We assessed 60 articles for full-text screening and 11 articles met our inclusion criteria and were included in this review. The PRISMA flow diagram for this study is shown in S1 Fig.

### Study characteristics

A total of 11 studies utilized an RCT study design with 2,666 patients have been included in this review. The studies were conducted in different countries and with different sample sizes, and settings (Table 1). The age range of the patients varied between 40 and 85 years, and the duration of mechanical ventilation ranged from 4 to 25 days. Seven of these studies were conducted in China, two in France and two in Spain.

### Quality assessment and risk of bias

The summary for the risk of bias of included studies is presented in S2 Fig. A total of seven studies underwent quality assessment, as four Chinese studies were available only as English

**Table 1. Characteristics of the included studies.**

| Study author | Study design | Study participants (N) | Country | Age (in years) | Reintubation Rate | Pre-extubation Oxygenation | Duration of MV | Time to reintubation | Adverse events | Effect summary |
|---|---|---|---|---|---|---|---|---|---|---|
| Guoqiang et al [18] | RCT, abstract | 44; 20 in the HFNC group and 24 in the NIPPV group | China | Between 40–85 | HFNC Vs. NIPPV [35.0% (7/20) vs. 4.2% (1/24), P < 0.05] | NR | NR | NR | NR | HFNC increases compliance and decrease complications, sequential NIPPV has a better final effect |
| Hernández et al [19] | RCT | 182; NIV (n = 92), HFNC (n = 90) | Spain | Mean age of participants (60) | 21 (23.3%) patients receiving NIV vs 35 (38.8%) of those receiving HFNC. (difference −15.5%; 95% confidence interval (CI) −28.3 to −1%) | Fraction of inspired oxygen (FiO2 SpO2 ≥ 92% in both groups | n ≥ 24 h | HFNC: Median 27 (IQR 8 to 48); NIV: 27 (6 to 47) | HFNC 0; NIV: 135 (42.9) | noninvasive ventilation with active humidification was superior to high-flow nasal cannula in preventing reintubation in adult critically ill patients at very high risk for extubation failure |
| Hernández et al. [20] | RCT | 604 | Spain | HFNC: 64.6 (15.4); NIV: 64.4 15.8) | (HFNC Vs. NIV), (22.8% Vs. 19.1%) | PaO2/FiO2 HFNC: 191 (34); NIV: 194 (37) | HFNC: 4 (2-9) days; NIV: 4 (2–8) days | HFNC: Median 26.5 (IQR 14 to 39); NIV: 21.5 (10 to 47) | HFNC 0; NIV: 135 (42.9) | The results showed that high-flow conditioned oxygen therapy was not inferior to NIV in preventing reintubation and postextubation respiratory failure. The study suggests that high-flow conditioned oxygen therapy may offer advantages over NIV for these patients. |
| Jing et al. [21] | RCT, abstract | HFNC (22) or NIV (20) | China | NR | NR | PaCO2, mmHg 50.5 (48–57.8) 53 (48.8–61.3) | NR | NR | HFNC 3 (18.1); NIV: 7 (30) | HFNC could be a potential alternative to NIV in weaning COPD patients with hypercapnia from invasive ventilation. The study found that at 3 hours and 24 hours after extubation, patients in the NIV group had lower pH levels and mean arterial pressure than those in the HFNC group |

(*Continued*)

**Table 1.** (Continued)

| Study author | Study design | Study participants (N) | Country | Age (in years) | Reintubation Rate | Pre-extubation Oxygenation | Duration of MV | Time to reintubation | Adverse events | Effect summary |
|---|---|---|---|---|---|---|---|---|---|---|
| **Tan et al** [22] | RCT | 96; 44 patients in the HFNC group and 42 patients in the NIV group | China | HFNC 68.4 ± 9.3 NIV 71.4 ± 7.8 | (HFNC Vs. NIV), (13.6% Vs. 14.3%) | PaCO2, mmHgHFNC = 50.5 (48–57.8), NIV = 53 (48.8–61.3) | NR | NR | NR | HFNC was non-inferior to NIV in preventing treatment failure and had better tolerance and comfort compared to NIV. HFNC also resulted in lower rates of treatment intolerance and fewer airway care interventions than NIV. |
| **Stéphan et al** [23] | RCT | 830; 416 BiPAP, 414 HFNC | France | HFNC: 63.8 (62.565.2); BiPAP: 63.9 (62.6–65.2) | (HFNC Vs. BiPAP, (14% Vs. 13.7%) | PaO2/FiO2 HFNC: 196 (187–204); BiPAP: 203 (195–212) | HFNC: 11.5 (5–25.4) hours; BiPAP: 13 (6.0–27.5) hours | NR | HFNC: 100 (24.1); BiPAP: 105 (% 25.2 ( | The study found that high-flow nasal oxygen therapy was not inferior to BiPAP, and the findings support the use of high-flow nasal oxygen therapy in similar patients. |
| **Xu & Liu** [24] | RCT | 100;50NIPPV,50HFNC | China | 68.39 ± 5.22 years | Control = 3 (6.00), Study = 1 (2.00) | NR | NR | NR | NR | The results showed that the overall response rate in the HFNC group was higher, and the group exhibited better respiratory function, diaphragmatic function, and comfort, with reduced dyspnea, fatigue, and sputum viscosity compared to the NIPPV group. |
| **Yang et al** [25] | RCT | treatment group (n = 40) and control group (n = 33) | China | Treatment = 69 ±9, control = 67±9 | NR | NR | NR | NR | NR | NIV combined with HFNCO was better than NIV alone in relieving diaphragm fatigue, promoting recovery of respiratory muscle strength, and shortening the average duration of NIV treatment time |

(*Continued*)

**Table 1.** (Continued)

| Study author | Study design | Study participants (N) | Country | Age (in years) | Reintubation Rate | Pre-extubation Oxygenation | Duration of MV | Time to reintubation | Adverse events | Effect summary |
|---|---|---|---|---|---|---|---|---|---|---|
| **Yu et al.** [26] | RCT, abstract | 72 HHFNC NPPV 36 patients in each group. | China | NR | NR | NR | NR | NR | NR | The results showed that HHFNC was more effective than NPPV in improving blood gas analysis index, respiratory rate, heart rate, mean arterial pressure, reintubation rate, incidence of tracheotomy, intensive care unit stay, incidence of adverse events, and mortality. |
| **Thille et al.** [27] | RCT | 651; 86 NIV high-flow nasal oxygen and 64 treated with high-flow nasal oxygen alone. | France | 70±10 | 11.8% vs 18.2% | Alone = 274 (93) NIV = 275 (89) | 5 days (interquartile range [IQR], 3-10); Allone = 5 (3–9); NIV = 6 (3-11) | NIV 33 hours (IQR, 7–81) and 39 hours (IQR, 12–67) with HFNC | No severe adverse events reported | The study concluded that the use of high-flow nasal oxygen with NIV after extubation can reduce the risk of reintubation in patients at high risk of extubation failure. |
| **Zhang et al.** [28] | RCT | 45; 21 cases in the highflow group), 24 cases in the non-invasive group | China | 64.5±5.3 vs 66.1±6.6 | NR | NR | NR | NR | NR | HFNCO is effective and safe in treating COPD patients after extubation and may be valuable for further clinical application. |
| | | | | | *NR: Not Reported* *MV: Mechanical ventilation* *RCT: Randomized Controlled Trial* *HFNC: High-Flow Nasal Cannula* *NIPPV: Non-Invasive Positive Pressure Ventilation* *NIV: Non-Invasive Ventilation* *PaO2/FiO2: Ratio of arterial partial pressure of oxygen to fraction of inspired oxygen* *PaCO2: Partial Pressure of Carbon Dioxide in Arterial Blood mmHg: Millimeters of Mercury* *HFNCO: High-Flow Nasal Cannula Oxygen* *BiPAP: bilevel positive airway pressure* | | | | | |

abstracts. Based on the Cochrane Risk of Bias Assessment Tool 2 (ROB2), the assessed studies were considered to have a low risk of bias, except for Xu et al. (2021) as shown in S2 Fig.

## Reintubation rate

The reintubation rates were reported in a few studies. Hernández et al. [19] found that the HFNC group had a reintubation rate of 38.8%, compared to 18.2% in the control group, and the NIV group had a reintubation rate of 23.3% versus 11.8% in the control group. In another

study by Hernández et al. [20], the baseline reintubation rates for both therapies were 20.0% to 25.0%, with a predefined noninferiority margin of 10% for the high flow group. After excluding respiratory-related reintubations, the difference in reintubation rates was 15.9% in the NIV group (50 patients) versus 16.9% in the high-flow group (49 patients). In the same study, 22.8% of patients (66 individuals) in the high-flow group and 19.1% (60 individuals) in the NIV group were reintubated.

## High-Flow Nasal Cannula Oxygen Therapy versus Non-Invasive Ventilation

The studies overall indicate that HFNC is comparable to NIV in preventing reintubation in critically ill adult patients following extubating. HFNC also offers advantages such as improved tolerance, comfort, and fewer complications related to airway care interventions when compared to NIV. However, the findings are not consistent across all studies, and some suggest that NIV with active humidification may be more effective than HFNC in preventing reintubation, particularly in patients at a high risk of extubating failure. For instance, Guoqiang et al. [18] reported that sequential NIPPV may yield better outcomes than sequential HFNC. Hernández et al. [19] found that NIV with active humidification was superior to HFNC in preventing reintubation in high-risk critically ill patients, although it may delay reintubation and prolong hospital stay. Jing et al. [21] observed that HFNC could be a potential alternative to NIV for weaning COPD patients from invasive ventilation, offering better comfort scores and fewer airway care interventions. Tan et al. [22] determined that HFNC was non-inferior to NIV in preventing treatment failure and demonstrated better tolerance and comfort. Stéphan F et al. [23] concluded that high-flow nasal oxygen therapy was not inferior to BiPAP, while Xu & Liu [24] found that HFNC sequential therapy was effective and safe in treating COPD patients with respiratory failure. Yang et al. [25] demonstrated that combining NIV with HFNC was superior to NIV alone in relieving diaphragm fatigue, promoting recovery of respiratory muscle strength, and shortening the average duration of NIV treatment. Yu et al. [26] found that HFNC was more effective than NPPV in improving various outcomes. Thille et al. [27] discovered that utilizing high-flow nasal oxygen with NIV after extubating reduced the risk of reintubation in high-risk patients. Finally, Zhang et al. [28] determined that HFNC is effective and safe in treating COPD patients following extubating and may hold clinical value, despite the NIV group exhibiting significantly higher oxygenation index than the HFNC group at 12 hours post-extubation.

## Discussion

Three methods, namely conventional oxygen therapy (COT), HFNC, and NIV, are available to improve oxygenation after extubation [29]. Guidelines suggest using NIV as a preventive measure for patients at risk of extubation failure, particularly those who are hypercapnic and obese. However, it remains unclear which patient subgroups may benefit more from HFNC or NIV, and the combined use of HFNC and NIV is now considered as a treatment option after extubation [30].

Traditionally, extubation failure risk has been defined as having at least one risk factor. However, this definition has limitations due to the presence of various high-risk factors in critically ill patients, such as age, prolonged mechanical ventilation, high Acute Physiology and Chronic Health Evaluation (APACHE) II scores, weaning difficulties, obesity, comorbidities, hypercapnia, airway problems, respiratory secretion management issues, heart failure, and chronic lung disease. Additionally, different risk factors have been identified for surgical patients. While existing meta-analyses have examined the effects of noninvasive ventilation

(NIV) post-extubation, there remains a paucity of literature specifically addressing the use of NIV in high-risk patients for extubation failure [31–33].

Compared to COT, HFNC has demonstrated increased chances of successful extubation in general critically ill patient populations and in low-risk extubation failure patients. It achieves this through flow-dependent positive end-expiratory pressure, more reliable inspired oxygen concentration, and improved heating and humidification efficiency. However, there is insufficient evidence to support its use in high-risk extubation failure patients [34–36]. Identifying patients who are likely to benefit the most from HFNC after extubation is crucial, given the increased resource demands in terms of equipment and staff [37, 38].

Recent research suggests that the immediate implementation of HFNC and NIV following extubation reduces the likelihood of post-extubation respiratory failure and reintubation compared to HFNC alone. Various RCTs have compared the effectiveness and safety of HFNC and NIV in weaning high-risk patients from invasive ventilation. Guoqiang et al. found that sequential noninvasive positive pressure ventilation (NIPPV) may yield better outcomes than sequential HFNC [18]. Hernández et al. indicated that NIV with active humidification was superior to HFNC in preventing reintubation, but it may increase hospital length of stay [19]. However, Frat et al. discovered that prophylactic NIV alternating with HFNC can reduce the risk of reintubation in COPD patients [12]. Tan et al. determined that HFNC was non-inferior to NIV in preventing treatment failure and offered better tolerance and comfort [22]. Other studies also found that combining NIV and HFNC can reduce the risk of reintubation in high-risk extubation failure patients [19–21, 23, 25–27].

The findings of our systematic review suggest that sequential NIV may have a better outcome compared to sequential HFNC, while NIV with active humidification was found to be superior to HFNC in preventing reintubation in critically ill patients at a very high risk of extubation failure [39].

Moreover, the results indicate that HFNC is more comfortable and tolerable for patients compared to NIV but may be less effective in preventing reintubation. However, a meta-analysis by Hua-Wei Huang including seven RCTs and 2781 patients found that HFNC had a similar reintubation rate compared to both COT and NIV, but exhibited a significantly lower reintubation rate than COT in critically ill patients [40]. Overall, HFNC and NIV have both demonstrated effectiveness in weaning high-risk patients from invasive ventilation, but their effectiveness may vary depending on the patient population and treatment strategies.

Furthermore, Feng et al. conducted a meta-analysis comprising eight studies with a total of 612 subjects, and found that HFNC and NIV had no statistically significant difference in reintubation rate among patients with hypercapnia, but NIV significantly reduced reintubation rates in patients without hypercapnia. HFNC also significantly reduced complication rates compared to NIV. No significant differences were observed between HFNC and NIV in terms of mortality and intensive care unit (ICU) length of stay [41]. HFNC assists hypercapnia by delivering a precise blend of warm, humidified oxygen at higher flow rates than traditional oxygen therapy. This facilitates better ventilation and clearance of carbon dioxide ($CO_2$) from the lungs, reducing elevated CO2 levels in the blood. HFNC's CPAP effect can also improve lung compliance and reduce the work of breathing, making it an effective non-invasive treatment option for patients with hypercapnia, especially in conditions like COPD exacerbations [42].

## Strengths and limitations

Our study possesses several notable strengths. First, it is a comprehensive and up-to-date systematic review that includes 11 RCTs, providing a robust evidence base for evaluating the

comparable efficacy of HFNC and NIV in high-risk patients. This systematic review is particularly strong due to its comprehensive review of the literature, enhancing the reliability and generalizability of our findings.

However, our study also has certain limitations that should be acknowledged. One limitation is the relatively small number of studies available for analysis, which may have resulted in overrepresentation of studies with low sample sizes and significant biases, and underrepresentation of RCTs with high quality of evidence. This restricts the interpretive scope of our results and may contribute to potential publication bias, despite our efforts to minimize it through comprehensive database searches. Nevertheless, this limitation serves as an indicator of the existing knowledge gap, highlighting the importance of further exploration by critical care researchers in this critical area. Another limitation is that the included studies exhibit significant heterogeneity in terms of design, population, and settings, which may introduce biases and affect the generalizability of the findings. In particular, the lack of reliable classification between AECOPD and non-AECOPD causes of hypercapnic respiratory failure. The lack of a rigorous analysis combining data to examine other confounding factors among the included patients in these studies further restricts the conclusions that can be drawn regarding this specific issue. Nonetheless, as with previous systematic reviews, high heterogeneity was expected and did not preclude this review This underscores the real-world diversity of clinical practices and patient populations, highlighting the need to improve quality and granularity of data Moreover, we acknowledge that we did not evaluate all the outcomes listed in the introduction section of the article, which could affect the comprehensiveness of our analysis. Additionally, we recognize that we have not explored study biases in detail, which is a critical aspect of systematic reviews.

## Conclusion

The current evidence does not provide a definitive answer regarding the use of HFNC as an alternative to NIV for previously intubated patients with COPD. Nonetheless, the findings of this systematic review provided valuable insights for clinical decision-making and guides the selection of appropriate respiratory support methods in this specific patient population.

Additional studies, including upcoming randomized non-inferiority experiments, are recommended to improve the accuracy of estimations in this area. The availability of different methods, including COT, HFNC, and NIV, offers clinicians various options for improving oxygenation after extubation. While NIV is also recommended as a preventive measure for patients at risk of extubation failure, the selection between HFNC and NIV remains uncertain. Combining HFNC and NIV shows promise, especially for high-risk patients. However, further research is needed to determine the optimal approach for different patient populations and to compare the effectiveness and outcomes of HFNC and NIV in specific subgroups.

## Supporting information

**S1 Fig. The PRISMA flow diagram.**
(TIF)

**S2 Fig. Risk of bias for included studies.**
(TIF)

**S1 Appendix. PICO framework.**
(DOCX)

## Author Contributions

**Conceptualization:** Ziyad F. Al Nufaiei.

**Data curation:** Ziyad F. Al Nufaiei, Raid M. Al Zhranei.

**Formal analysis:** Ziyad F. Al Nufaiei.

**Investigation:** Ziyad F. Al Nufaiei.

**Methodology:** Ziyad F. Al Nufaiei.

**Software:** Ziyad F. Al Nufaiei.

**Supervision:** Ziyad F. Al Nufaiei.

**Validation:** Ziyad F. Al Nufaiei, Raid M. Al Zhranei.

**Visualization:** Ziyad F. Al Nufaiei.

**Writing – original draft:** Ziyad F. Al Nufaiei, Raid M. Al Zhranei.

**Writing – review & editing:** Raid M. Al Zhranei.

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
