## [Decision Letter · Decision Letter 0]

2 Aug 2023

PONE-D-23-20803High-Flow Nasal Cannula Oxygen Therapy versus Non-Invasive Ventilation in Patients at very high risk for extubation failure: A Systematic Review of Randomized Controlled TrialsPLOS ONE

Dear Dr. AL NUFAIEI,

Thank you for submitting your manuscript to PLOS ONE. After careful consideration, we feel that it has merit but does not fully meet PLOS ONE’s publication criteria as it currently stands. Therefore, we invite you to submit a revised version of the manuscript that addresses the points raised during the review process

We look forward to receiving your revised manuscript.

Kind regards,

Kartikeya Rajdev, MD

Academic Editor

PLOS ONE

Journal Requirements:

3. Please ensure that you include a title page within your main document. You should list all authors and all affiliations as per our author instructions and clearly indicate the corresponding author.

5. We note that this manuscript is a systematic review or meta-analysis; our author guidelines therefore require that you use PRISMA guidance to help improve reporting quality of this type of study. Please upload copies of the completed PRISMA checklist as Supporting Information with a file name “PRISMA checklist”.

Reviewers' comments:

Reviewer's Responses to Questions

**Comments to the Author**

1. Is the manuscript technically sound, and do the data support the conclusions?

Reviewer #1: Yes

Reviewer #2: No

2. Has the statistical analysis been performed appropriately and rigorously? 

Reviewer #1: Yes

Reviewer #2: N/A

3. Have the authors made all data underlying the findings in their manuscript fully available?

Reviewer #1: Yes

Reviewer #2: Yes

4. Is the manuscript presented in an intelligible fashion and written in standard English?

Reviewer #1: Yes

Reviewer #2: No

5. Review Comments to the Author

Reviewer #1: Well written systematic review and meta analysis of RCT comparing efficacy of NIV vs HFNC in recently extubated patients. It’s a very interesting and important topic and of great importance in current clinical practice . As mentioned HFNC is extensively used in hospitals and much preferred by patients because of convenience and comfort to use.

Points that could have included on discussion:

How HFNC helps in hypercapnia

The study still leave the readers in an inconclusive state that which is more beneficial. I think it’s better to conclude that a combination of both is more effective that one Alone .

Reviewer #2: Abstract:

Background: Line 2 in background section I suggest changing “them” to patients

Line 3, I suggest changing “some” to few or give an appropriate measure of this problem. Some patients do not respond well to it, is very generic.

Methods: Line 5 please clearly write length of ICU stay or length of hospital stay?

Results: Line 1 and 2, please indicate did you only select RCTs of COPD patients with high risk of reintubation or all RCTs of patients with high risk for re-intubation? In the background section line 5 you have mentioned you aimed to study HFNC versus NIV among previously intubated COPD patients.

Introduction: Line 1 and 2, Ref [1] this was a pilot study done in 10 COPD patients to see hemodynamic effects of spontaneous breathing trial on pulmonary artery occlusion pressure. Patients were treated with dobutamine infusion followed by Levosemindan to assess the hemodynamic benefit of both infusions. Reference provided here does not match by statement made by authors. I suggest providing a different reference for the statement made in line one and two or no reference needed for this statement.

Line 4- I will suggest changing “deteriorating respiratory mechanics” to deteriorating hemodynamics as study cited here studied hemodynamic effects of spontaneous breathing trials in COPD patients. This study did not evaluate respiratory mechanics.

Line 5-6 Ref [2], this study evaluated the attributable morbidity and mortality of ventilator-associated pneumonia (VAP) in intensive care unit (ICU) patients. Findings noted increase ICU length of stay and trend towards increased mortality with VAP. This study did not evaluate adverse effects of prolonged mechanical ventilation. This reference is not the most appropriate to reflect the statement made here. Please consider changing reference, suggested reference would be: Coplin WM, Pierson DJ, Cooley KD, et al. Implications of extubation delay in brain-injured patients meeting standard weaning criteria. Am J Respir Crit Care Med 2000;161:1530-6.

Line 9, 10, 11: Ref [5], this study evaluated home NIV monitoring in COPD patients, not a appropriate reference for NIV patients immediately post extubation.

Ref [6] appropriately cited and supports the statement made by authors.

Line 13, 14, 15 Ref [7], this study evaluated utility of Heliox in acute COPD exacerbation. Results noted Heliox improves respiratory acidosis, encephalopathy, and the respiratory rate more quickly than air/O2 but does not prevent NIV failure. I suggest using a more appropriate reference to support the statement made here.

Ref [8] Multinational observational cohort study recording trends of mechanical ventilation from 1998 to 2004 does not support statement made here. Please consider adding another reference supporting statement made.

Line 15 and 16, NIV associated with difficulty is aspiration is an incorrect statement. Data supports NIV associated with increased aspiration risk. Please make appropriate corrections here. Ref [9] appropriately cited.

Ref [10] excellent reference to support the use of high flow oxygen in acute hypoxemic respiratory failure. FLORALI RCT studying high flow oxygen with standard of care and NIV in acute hypoxemic respiratory failure (non-hypercapnic patients). High flow oxygen group showed significantly lower ventilator free days and increased 90-day survival.

Methods:

Inclusion criteria:

Line 2 and 3 please make it clear if you only studied RCTs with COPD patients?

Search Strategy: Line 3 mentions appendix II details the search term but there is no appendix shown in manuscript.

Screening, data extraction and quality assessment: Line 9 Ref [20] listed as type of biases but this is a study evaluating HFNC versus NIPPV outcomes post extubation. Please cite this reference accordingly.

Results: Line 3 you have mentioned 30 articles were removed as you were unable to retrieve those articles. Please explain this in more detail as important studies may have been missed in this analysis (risk for bias in analysis).

Study characteristics: Line 5, you have mentioned full texts were not available for 3 abstracts (18), (21), and (26). Do these numbers mean references listed in manuscript or these numbers reflect something else? Ref 18 is a study on assessing biases in RTCs. Ref 21 and 26 are open access articles easily available. Please explain?

Table 1: There is significant heterogeneity in studies listed in this table. I will suggest authors make a separate column to indicate potential weaknesses or biases of individual studies mentioned in table.

Please also consider adding columns in your table including adverse events, and time to reintubation as you have listed these as your study outcomes.

Ref [20], study from China reported the rate of tracheal re-intubation within 7 days in the HFNC group was significantly higher than that in the NIPPV group [35.0% (7/20) vs. 4.2 % (1/24), P < 0.05]. You have mentioned re-intubation rate were not reported in this study. Please make corrections in your table.

Table 1 Ref [21], column 6 the reintubation needs correction. In this study, 21 [23.3%] patients NIV group vs 35 [38.8%] patients HFNC group. I would suggest making this change to the table. Too many numbers in this column are confusing.

Table 1 Ref [22] column 6 please indicate re-intubation rates in both groups (HFNC vs NIV). In this study included total 604 patients, 66 patients (22.8%) in the high-flow group vs 60 (19.1%) in the NIV group required re-intubation.

Table 1 Ref [23] small pilot studies, interpretation is limited from this RCT. Data showing in abstract reflects only 48 hours post extubation.

Table 1 Ref [24] column 6 please indicate re-intubation rates were similar in both groups. In this study, 6(13.6) patients in HFNC group and 6(14.29) patients in NIV group required re-intubation. This was shown in table 2 of this article.

Table 1 Ref [25] column 6 needs correction. In this French study, reintubation was performed in 57 patients with BiPAP (13.7%) and 58withhigh-flownasal oxygen therapy (14.0%) (P = .99).

Table 1 reference [27] this is a meta-analysis evaluating HFNC in type II respiratory failure. I would suggest removing this meta-analysis from your systematic review. This did not evaluate the utility of HFNC post extubation. This study doesn’t match with your inclusion criteria please explain why study was included in analysis.

Table 1 [Ref 29] agree with authors.

Quality assessment and risk of bias:

Reintubation: Line 1 (19) please explain what this number indicates as findings written in line 1-4 are Ref [21]. Line 1-4 need corrections.

Line 4 (20) please explain what this number indicates as study cited here is not Ref [20]. Please cite reference at the right place in the article.

High-Flow Nasal Cannula Oxygen Therapy versus Non-Invasive Ventilation

Multiple mistakes in citing the reference. Reference number cited does not match with reference listed. Please make these corrections so this section can be reviewed appropriately.

Discussion:

Paragraph 2, Line 6 and 7 mentioned: To date, o meta-analysis or systematic review specifically focused on the role of NIV post extubation. This statement made here by authors does not support the existing literature. You have cited the following meta-analysis and systematic review at Ref [46] contradicting statement made here.

Huang HW, Sun XM, Shi ZH, et al. Effect of High-Flow Nasal Cannula Oxygen Therapy Versus Conventional Oxygen Therapy and Noninvasive Ventilation on Reintubation Rate in Adult Patients After Extubation: A Systematic Review and Meta-Analysis of Randomized Controlled Trials. J Intensive Care Med. 2018;33(11):609-623. doi:10.1177/0885066617705118

Please review references cited in the discussion section and references listed. There are multiple mismatches in listings.

Strengths and limitations:

Comprehensive review of literature is the strength of this systematic review.

Weaknesses authors did not evaluate all the outcomes listed in the introduction section of the article. There is significant heterogeneity in the studies listed in this systematic review. Authors have not explored study biases in this systematic review. Studies with low sample size and significant biases may have been overrepresented in this systematic review. RCTs with high quality of evidence may have been underrepresented.

References:

Ref [1] Inappropriately cited

Ref [2] his study evaluated the attributable morbidity and mortality of ventilator-associated pneumonia (VAP) in intensive care unit (ICU) patients . Findings noted increase ICU length of stay and trend towards increased mortality with VAP. This reference is not the most appropriate to reflect the statement made by authors.

Ref [15] and [16] are listed in the reference list but not cited anywhere in the manuscript.

6. PLOS authors have the option to publish the peer review history of their article (what does this mean?). If published, this will include your full peer review and any attached files.

Reviewer #1: **Yes: **Gisha Mohan

Reviewer #2: **Yes: **Rajendra K Karnatak

---

## [Author Response · Author response to Decision Letter 0]

31 Oct 2023

Thank you for your time.

I have went through all your comments one by one. I have attached my response to your comments in the attachment section

---

## [Editor Report · Decision Letter 1]

13 Nov 2023

PONE-D-23-20803R1High-Flow Nasal Cannula Oxygen Therapy versus Non-Invasive Ventilation in Patients at very high risk for extubation failure: A Systematic Review of Randomized Controlled TrialsPLOS ONE

Dear Dr. AL NUFAIEI,

Thank you for submitting your manuscript to PLOS ONE. After careful consideration, we feel that it has merit but does not fully meet PLOS ONE’s publication criteria as it currently stands. Therefore, we invite you to submit a revised version of the manuscript that addresses the points raised during the review process.

We look forward to receiving your revised manuscript.

Kind regards,

Kartikeya Rajdev, MD

Academic Editor

PLOS ONE

**Additional Editor Comments:**

The article is being returned to the author(s) so that the revised version of the article can be uploaded by the author(s). The current uploaded 'revised' document is the same as the original version, therefore the reviewers can not see the revised version.

Reviewers' comments:

Cannot see any changes made. 

---

## [Author Response · Author response to Decision Letter 1]

13 Dec 2023

Many thanks for the opportunity to revise and re-submit this manuscript. We have addressed all the reviewers’ comments and outlined our responses and the amendments made. We thank the reviewers for noting the importance of this manuscript. We hope that you and the reviewers will feel we have addressed the valuable input we have received and look forward to hearing from you regarding publishing our manuscript

---

## [Decision Letter · Decision Letter 2]

2 Jan 2024

PONE-D-23-20803R2High-Flow Nasal Cannula Oxygen Therapy versus Non-Invasive Ventilation in Patients at very high risk for extubation failure: A Systematic Review of Randomized Controlled TrialsPLOS ONE

Dear Dr. AL NUFAIEI,

Thank you for submitting your manuscript to PLOS ONE. After careful consideration, we feel that it has merit but does not fully meet PLOS ONE’s publication criteria as it currently stands. Therefore, we invite you to submit a revised version of the manuscript that addresses the points raised during the review process.

We look forward to receiving your revised manuscript.

Kind regards,

Kartikeya Rajdev, MD

Academic Editor

PLOS ONE

Journal Requirements:

Reviewers' comments:

Reviewer's Responses to Questions

**Comments to the Author**

1. If the authors have adequately addressed your comments raised in a previous round of review and you feel that this manuscript is now acceptable for publication, you may indicate that here to bypass the “Comments to the Author” section, enter your conflict of interest statement in the “Confidential to Editor” section, and submit your "Accept" recommendation.

Reviewer #2: All comments have been addressed

2. Is the manuscript technically sound, and do the data support the conclusions?

Reviewer #2: Yes

3. Has the statistical analysis been performed appropriately and rigorously? 

Reviewer #2: Yes

4. Have the authors made all data underlying the findings in their manuscript fully available?

Reviewer #2: Yes

5. Is the manuscript presented in an intelligible fashion and written in standard English?

Reviewer #2: Yes

6. Review Comments to the Author

Reviewer #2: This manuscript need minor corrections. Please make corrections as suggested below

Re-review

High-Flow Nasal Cannula Oxygen Therapy versus Non-Invasive Ventilation in Patients at very high risk for extubation failure: A Systematic Review of Randomized Controlled Trials

Abstract:

Background: Line 2 in background section I suggest changing “them” to patients - Corrections made by authors

Line 3, I suggest changing “some” to few or give an appropriate measure of this problem. Some patients do not respond well to it, is very generic. – Corrections made by authors

Methods: Line 3, please indicate did you only select RCTs of COPD patients with high risk of reintubation or all RCTs of patients with high risk for re-intubation? In the background section line 5 you have mentioned you aimed to study HFNC versus NIV among previously intubated COPD patients. Corrections made by authors

Line 5 please clearly write length of ICU stay or length of hospital stay? Corrections made by authors

Introduction: Line 1 and 2, Ref [1] this was a pilot study done in 10 COPD patients to see hemodynamic effects of spontaneous breathing trial on pulmonary artery occlusion pressure. Patients were treated with dobutamine infusion followed by Levosemindan to assess the hemodynamic benefit of both infusions. Reference provided here does not match by statement made by authors. I suggest providing a different reference for the statement made in line one and two or no reference needed for this statement. Corrections made by authors now added reference 1,2,3 this will support the statement made here.

Line 4- I will suggest changing “deteriorating respiratory mechanics” to deteriorating hemodynamics as study cited here studied hemodynamic effects of spontaneous breathing trials in COPD patients. This study did not evaluate respiratory mechanics. Corrections made by authors reference provided here does show effect of extubation deteriorating hemodynamics.

Line 5-6 Ref [2], this study evaluated the attributable morbidity and mortality of ventilator-associated pneumonia (VAP) in intensive care unit (ICU) patients. Findings noted increase ICU length of stay and trend towards increased mortality with VAP. This study did not evaluate adverse effects of prolonged mechanical ventilation. This reference is not the most appropriate to reflect the statement made here. Please consider changing reference, suggested reference would be: Coplin WM, Pierson DJ, Cooley KD, et al. Implications of extubation delay in brain-injured patients meeting standard weaning criteria. Am J Respir Crit Care Med 2000;161:1530-6.

Corrections made by authors

Line 9, 10, 11: Ref [5], this study evaluated home NIV monitoring in COPD patients, not a appropriate reference for NIV patients immediately post extubation. This reference is corrected by authors.

Ref [6] appropriately cited and supports the statement made by authors.

Line 13, 14, 15 Ref [7], this study evaluated utility of Heliox in acute COPD exacerbation. Results noted Heliox improves respiratory acidosis, encephalopathy, and the respiratory rate more quickly than air/O2 but does not prevent NIV failure. I suggest using a more appropriate reference to support the statement made here. Correction made by authors new reference [8] added

Ref [8] Multinational observational cohort study recording trends of mechanical ventilation from 1998 to 2004 does not support statement made here. Please consider adding another reference supporting statement made.

Line 15 and 16, NIV associated with difficulty is aspiration is an incorrect statement. Data supports NIV associated with increased aspiration risk. Please make appropriate corrections here.

Ref [9] appropriately cited.

Ref [10] excellent reference to support the use of high flow oxygen in acute hypoxemic respiratory failure. FLORALI RCT studying high flow oxygen with standard of care and NIV in acute hypoxemic respiratory failure (non-hypercapnic patients). High flow oxygen group showed significantly lower ventilator free days and increased 90-day survival. No correction needed here.

Methods:

Inclusion criteria:

Search Strategy: Line 3 mentions appendix II details the search term but there is no appendix shown in manuscript. Corrections made here.

Screening, data extraction and quality assessment: Line 9 Ref [20] listed as type of biases but this is a study evaluating HFNC versus NIPPV outcomes post extubation. Please cite this reference accordingly. Reference corrected by authors.

Results: Line 3 you have mentioned 30 articles were removed as you were unable to retrieve those articles. Please explain this in more detail as important studies may have been missed in this analysis (risk for bias in analysis). Corrections made by authors.

Study characteristics: Line 5, you have mentioned full texts were not available for 3 abstracts (18), (21), and (26). Do these numbers mean references listed in manuscript or these numbers reflect something else? Ref 18 is a study on assessing biases in RTCs. Ref 21 and 26 are open access articles easily available. Please explain? Corrections made by authors

Ref [20], study from China reported the rate of tracheal re-intubation within 7 days in the HFNC group was significantly higher than that in the NIPPV group [35.0% (7/20) vs. 4.2 % (1/24), P < 0.05]. You have mentioned re-intubation rate were not reported in this study. Please make corrections in your table. Authors have removed this study from the table and analysis. Corrections made by authors

Table 1 Ref [21], column 6 the reintubation needs correction. Authors have made these corrections.

Table 1 Ref [22] column 6 please indicate re-intubation rates in both groups (HFNC vs NIV). In this study included total 604 patients, 66 patients (22.8%) in the high-flow group vs 60 (19.1%) in the NIV group required re-intubation. Authors have changed this reference to reference [23] now and numbers are appropriately reflected.

Table 1 New Ref [22] in this study total 182 patients, 92 received NIV and 90 HFNC. Reintubation was required in 21 (23.3%) patients receiving NIV vs 35 (38.8%) of those receiving HFNC (difference −15.5%; 95% confidence interval (CI) −28.3 to −1%). This needs to be correctly reflected in the table please make these corrections. Authors have changed this reference to reference [23] now and numbers are appropriately reflected. Corrections made by authors

Table 1 Ref [24] Jing et al. small pilot studies, interpretation is limited from this RCT. Data showing in abstract reflects only 48 hours post extubation.

Table 1 Ref [25] column 6 please indicate re-intubation rates were similar in both groups. In this study, 6(13.6) patients in HFNC group and 6(14.29) patients in NIV group required re-intubation. This was shown in table 2 of this article. Authors made these corrections.

Table 1 Ref [26] column 6 needs correction. In this French study, reintubation was performed in 57 patients with BiPAP (13.7%) and 58 with high-flow nasal oxygen therapy (14.0%) (P = .99). Authors made these corrections.

Table 1 reference [27] this is a meta-analysis evaluating HFNC in type II respiratory failure. I would suggest removing this meta-analysis from your systematic review. This did not evaluate the utility of HFNC post extubation. This study doesn’t match with your inclusion criteria please explain why study was included in analysis. Authors made corrections.

Table 1 [Ref 29] agree with authors.

Quality assessment and risk of bias:

Reintubation: Line 1 (19) please explain what this number indicates as findings written in line 1-4 are Ref [21]. Line 1-4 need corrections. Line 4 (20) please explain what this number indicates as study cited here is not Ref [20]. Please cite reference at the right place in the article. This has been corrected by authors.

High-Flow Nasal Cannula Oxygen Therapy versus Non-Invasive Ventilation

Multiple mistakes in citing the reference. Reference number cited does not match with reference listed. Please make these corrections so this section can be reviewed appropriately. Authors have corrected this.

Discussion:

Paragraph 2, Line 6 and 7 mentioned: To date, o meta-analysis or systematic review specifically focused on the role of NIV post extubation. This statement made here by authors does not support the existing literature. You have cited the following meta-analysis and systematic review at Ref [46] contradicting statement made here.

Huang HW, Sun XM, Shi ZH, et al. Effect of High-Flow Nasal Cannula Oxygen Therapy Versus Conventional Oxygen Therapy and Noninvasive Ventilation on Reintubation Rate in Adult Patients After Extubation: A Systematic Review and Meta-Analysis of Randomized Controlled Trials. J Intensive Care Med. 2018;33(11):609-623. doi:10.1177/0885066617705118

Please review references cited in the discussion section and references listed. There are multiple mismatches in listings. Authors have made this correction.

Strengths and limitations:

Strength: Comprehensive review of literature is the strength of this systematic review.

Weaknesses: There is significant heterogeneity in the studies listed in this systematic review. Authors have not explored study biases in this systematic review. Studies with low sample size and significant biases may have been overrepresented in this systematic review. RCTs with high quality of evidence may have been underrepresented.

References:

References:

Ref [15] and [16] not cited in the article but are listed in references. Authors have made these corrections.

7. PLOS authors have the option to publish the peer review history of their article (what does this mean?). If published, this will include your full peer review and any attached files.

Reviewer #2: **Yes: **Rajendra Karnatak

---

## [Author Response · Author response to Decision Letter 2]

31 Jan 2024

Many thanks for the opportunity to revise and re-submit this manuscript. We have now addressed the most recent points raised during the review process and outlined below our responses and the amendments made. We thank you for noting the importance of this manuscript and the opportunity to publish in your respective journal. We hope that you and the reviewers will feel we have addressed the valuable input we have received and look forward to hearing from you regarding publishing our manuscript.

---

## [Decision Letter · Decision Letter 3]

15 Feb 2024

High-Flow Nasal Cannula Oxygen Therapy versus Non-Invasive Ventilation in Patients at very high risk for extubation failure: A Systematic Review of Randomized Controlled Trials

PONE-D-23-20803R3

Dear Dr. Al Nufaiei,

We’re pleased to inform you that your manuscript has been judged scientifically suitable for publication and will be formally accepted for publication once it meets all outstanding technical requirements.

Kind regards,

Kartikeya Rajdev, MD

Academic Editor

PLOS ONE

Additional Editor Comments (optional):

Reviewers' comments:

Reviewer's Responses to Questions

**Comments to the Author**

1. If the authors have adequately addressed your comments raised in a previous round of review and you feel that this manuscript is now acceptable for publication, you may indicate that here to bypass the “Comments to the Author” section, enter your conflict of interest statement in the “Confidential to Editor” section, and submit your "Accept" recommendation.

Reviewer #2: All comments have been addressed

2. Is the manuscript technically sound, and do the data support the conclusions?

Reviewer #2: Yes

3. Has the statistical analysis been performed appropriately and rigorously? 

Reviewer #2: Yes

4. Have the authors made all data underlying the findings in their manuscript fully available?

Reviewer #2: Yes

5. Is the manuscript presented in an intelligible fashion and written in standard English?

Reviewer #2: Yes

6. Review Comments to the Author

Reviewer #2: Re-review

High-Flow Nasal Cannula Oxygen Therapy versus Non-Invasive Ventilation in Patients at very high risk for extubation failure: A Systematic Review of Randomized Controlled Trials

Abstract:

Background: Line 2 in background section I suggest changing “them” to patients - Corrections made by authors

Line 3, I suggest changing “some” to few or give an appropriate measure of this problem. Some patients do not respond well to it, is very generic. – Corrections made by authors

Methods: Line 3, please indicate did you only select RCTs of COPD patients with high risk of reintubation or all RCTs of patients with high risk for re-intubation? In the background section line 5 you have mentioned you aimed to study HFNC versus NIV among previously intubated COPD patients. Corrections made by authors

Line 5 please clearly write length of ICU stay or length of hospital stay? Corrections made by authors

Introduction: Line 1 and 2, Ref [1] this was a pilot study done in 10 COPD patients to see hemodynamic effects of spontaneous breathing trial on pulmonary artery occlusion pressure. Patients were treated with dobutamine infusion followed by Levosemindan to assess the hemodynamic benefit of both infusions. Reference provided here does not match by statement made by authors. I suggest providing a different reference for the statement made in line one and two or no reference needed for this statement. Corrections made by authors now added reference 1,2,3 this will support the statement made here.

Line 4- I will suggest changing “deteriorating respiratory mechanics” to deteriorating hemodynamics as study cited here studied hemodynamic effects of spontaneous breathing trials in COPD patients. This study did not evaluate respiratory mechanics. Corrections made by authors reference provided here does show effect of extubation deteriorating hemodynamics.

Line 5-6 Ref [2], this study evaluated the attributable morbidity and mortality of ventilator-associated pneumonia (VAP) in intensive care unit (ICU) patients. Findings noted increase ICU length of stay and trend towards increased mortality with VAP. This study did not evaluate adverse effects of prolonged mechanical ventilation. This reference is not the most appropriate to reflect the statement made here. Please consider changing reference, suggested reference would be: Coplin WM, Pierson DJ, Cooley KD, et al. Implications of extubation delay in brain-injured patients meeting standard weaning criteria. Am J Respir Crit Care Med 2000;161:1530-6.

Corrections made by authors

Line 9, 10, 11: Ref [5], this study evaluated home NIV monitoring in COPD patients, not a appropriate reference for NIV patients immediately post extubation. This reference is corrected by authors.

Ref [6] appropriately cited and supports the statement made by authors.

Line 13, 14, 15 Ref [7], this study evaluated utility of Heliox in acute COPD exacerbation. Results noted Heliox improves respiratory acidosis, encephalopathy, and the respiratory rate more quickly than air/O2 but does not prevent NIV failure. I suggest using a more appropriate reference to support the statement made here. Correction made by authors new reference [8] added

Ref [8] Multinational observational cohort study recording trends of mechanical ventilation from 1998 to 2004 does not support statement made here. Please consider adding another reference supporting statement made.

Line 15 and 16, NIV associated with difficulty is aspiration is an incorrect statement. Data supports NIV associated with increased aspiration risk. Please make appropriate corrections here.

Ref [9] appropriately cited.

Ref [10] excellent reference to support the use of high flow oxygen in acute hypoxemic respiratory failure. FLORALI RCT studying high flow oxygen with standard of care and NIV in acute hypoxemic respiratory failure (non-hypercapnic patients). High flow oxygen group showed significantly lower ventilator free days and increased 90-day survival.

Methods:

Inclusion criteria:

Search Strategy: Line 3 mentions appendix II details the search term but there is no appendix shown in manuscript. Corrections made here.

Screening, data extraction and quality assessment: Line 9 Ref [20] listed as type of biases but this is a study evaluating HFNC versus NIPPV outcomes post extubation. Please cite this reference accordingly. Reference corrected by authors.

Results: Line 3 you have mentioned 30 articles were removed as you were unable to retrieve those articles. Please explain this in more detail as important studies may have been missed in this analysis (risk for bias in analysis). Corrections made by authors.

Study characteristics: Line 5, you have mentioned full texts were not available for 3 abstracts (18), (21), and (26). Do these numbers mean references listed in manuscript or these numbers reflect something else? Ref 18 is a study on assessing biases in RTCs. Ref 21 and 26 are open access articles easily available. Please explain?

Table 1: There is significant heterogeneity in studies listed in this table. I will suggest authors make a separate column to indicate potential weaknesses or biases of individual studies mentioned in table.

Please also consider adding columns in your table including adverse events, and time to reintubation as you have listed these as your study outcomes.

Ref [20], study from China reported the rate of tracheal re-intubation within 7 days in the HFNC group was significantly higher than that in the NIPPV group [35.0% (7/20) vs. 4.2 % (1/24), P < 0.05]. You have mentioned re-intubation rate were not reported in this study. Please make corrections in your table. Authors have removed this study from the table and analysis.

Table 1 Ref [21], column 6 the reintubation needs correction. Authors have made these corrections.

Table 1 Ref [22] column 6 please indicate re-intubation rates in both groups (HFNC vs NIV). In this study included total 604 patients, 66 patients (22.8%) in the high-flow group vs 60 (19.1%) in the NIV group required re-intubation. Authors have changed this reference to reference [23] now and numbers are appropriately reflected.

Table 1 New Ref [22] in this study total 182 patients, 92 received NIV and 90 HFNC. Reintubation was required in 21 (23.3%) patients receiving NIV vs 35 (38.8%) of those receiving HFNC (difference −15.5%; 95% confidence interval (CI) −28.3 to −1%). This needs to be correctly reflected in the table please make these corrections.

Authors have changed this reference to reference [23] now and numbers are appropriately reflected.

Table 1 Ref [24] Jing et al. small pilot studies, interpretation is limited from this RCT. Data showing in abstract reflects only 48 hours post extubation.

Table 1 Ref [25] column 6 please indicate re-intubation rates were similar in both groups. In this study, 6(13.6) patients in HFNC group and 6(14.29) patients in NIV group required re-intubation. This was shown in table 2 of this article. Authors made these corrections.

Table 1 Ref [26] column 6 needs correction. In this French study, reintubation was performed in 57 patients with BiPAP (13.7%) and 58 with high-flow nasal oxygen therapy (14.0%) (P = .99). Authors made these corrections.

Table 1 reference [27] this is a meta-analysis evaluating HFNC in type II respiratory failure. I would suggest removing this meta-analysis from your systematic review. This did not evaluate the utility of HFNC post extubation. This study doesn’t match with your inclusion criteria please explain why study was included in analysis. Authors made corrections.

Table 1 [Ref 29] agree with authors.

Quality assessment and risk of bias:

Reintubation: Line 1 (19) please explain what this number indicates as findings written in line 1-4 are Ref [21]. Line 1-4 need corrections.

Line 4 (20) please explain what this number indicates as study cited here is not Ref [20]. Please cite reference at the right place in the article. This has been corrected.

High-Flow Nasal Cannula Oxygen Therapy versus Non-Invasive Ventilation

Multiple mistakes in citing the reference. Reference number cited does not match with reference listed. Please make these corrections so this section can be reviewed appropriately.

Discussion:

Paragraph 2, Line 6 and 7 mentioned: To date, o meta-analysis or systematic review specifically focused on the role of NIV post extubation. This statement made here by authors does not support the existing literature. You have cited the following meta-analysis and systematic review at Ref [46] contradicting statement made here.

Huang HW, Sun XM, Shi ZH, et al. Effect of High-Flow Nasal Cannula Oxygen Therapy Versus Conventional Oxygen Therapy and Noninvasive Ventilation on Reintubation Rate in Adult Patients After Extubation: A Systematic Review and Meta-Analysis of Randomized Controlled Trials. J Intensive Care Med. 2018;33(11):609-623. doi:10.1177/0885066617705118

Please review references cited in the discussion section and references listed. There are multiple mismatches in listings. Authors have made this correction.

Strengths and limitations:

Strength: Comprehensive review of literature is the strength of this systematic review.

Weaknesses: There is significant heterogeneity in the studies listed in this systematic review. Authors have not explored study biases in this systematic review. Studies with low sample size and significant biases may have been overrepresented in this systematic review. RCTs with high quality of evidence may have been underrepresented.

References:

References:

Ref [15] and [16] not cited in the article but are listed in references. Authors have made these corrections.

7. PLOS authors have the option to publish the peer review history of their article (what does this mean?). If published, this will include your full peer review and any attached files.

Reviewer #2: **Yes: **Rajendra Karnatak

---

## [Editor Report · Acceptance letter]

25 Mar 2024

PONE-D-23-20803R3 

PLOS ONE

Dear Dr. Al Nufaiei, 

I'm pleased to inform you that your manuscript has been deemed suitable for publication in PLOS ONE. Congratulations! Your manuscript is now being handed over to our production team.

Kind regards, 

on behalf of

Dr. Kartikeya Rajdev 

Academic Editor

PLOS ONE